# A retrospective cohort study assessing medication coverage in patients with prostate cancer prescribed luteinizing hormone releasing hormone (LHRH) agonists in England

Ian Sayers[1]*, Sara Joao Carvalho[2], Jennifer Davidson[2], Naomi Elster[3], Rakesh Heer[4], Mohammad Raja[5], Kate Higgs[5], Andrew Nolan[5], Jelena Sassmann[5]

1 Royal Wolverhampton Hospitals NHS Trust, Wolverhampton, United Kingdom, 2 CorEvitas, London, United Kingdom, 3 Prostate Cancer Research, London, United Kingdom, 4 Imperial College London, London, United Kingdom, 5 Ipsen, London, United Kingdom

* igsayers8@outlook.com

## Abstract

### Purpose

This study aims to assess adherence to luteinising hormone-releasing hormone (LHRH) agonist treatment for prostate cancer (PC) in England, considering formulation-related differences, their impact on overall survival, and the association with changes in prostate-specific antigen (PSA) levels over time.

### Methods

In this retrospective cohort study, utilising primary care data from the Clinical Practice Research Datalink (CPRD) Aurum database linked to Hospital Episode Statistics (HES) and Office for National Statistics (ONS) death registrations, we assessed male patients aged 40 and above diagnosed with PC and prescribed 1-, 3-, or 6-monthly LHRH agonist injections between January 2007 and December 2019. The primary objectives were to measure adherence through proportion of days covered (PDC) and characterize delayed injections, while secondary objectives included assessment of patient demographics, comorbidities, overall survival, and PSA levels. Descriptive statistics were employed, with follow-up restricted to one year for PSA and testosterone measurements due to data availability constraints.

### Results

The study included 32,777 patients with PC receiving LHRH agonists. Most patients (67%) were prescribed 3-monthly formulations, while only 2% received 6-monthly formulations. The mean age of the study population was 74.1 years. Over 80% of patients had at least one comorbidity, with hypertension being the most common. 94% of patients

**Data availability statement:** Data cannot be shared publicly because it is person-level routinely collected medical data subject to data governance and privacy restrictions. The data are available upon request from the Clinical Practice Research Datalink (CPRD) and this includes linkage to Hospital Episode Statistics (HES) data. Access to these data requires the purchase of a license, and the terms of this license do not permit the authors to make the data publicly available. Licenses are available directly from CPRD (http://www.cprd.com): The Clinical Practice Research Datalink Group, The Medicines and Healthcare products Regulatory Agency, 10 South Colonnade, Canary Wharf, London E14 4PU. The data utilised in this research are classified as third-party data, meaning they are not owned or directly collected by the authors. The authors did not possess any special access privileges that would grant them capabilities beyond those available to other researchers and it is confirmed that these data are accessible to others in the same manner as they were accessed by the authors. Information regarding data access and data linkage is available from https://www.cprd.com/data-accessand, https://www.cprd.com/cprd-linked-data.

**Funding:** This study was funded by Ipsen, UK. The funder was involved in the design of the study, analysis and interpretation of the data, and development of the manuscript.

**Competing interests:** Sara Joao Carvalho and Jennifer Davidson are employed by CorEvitas, an organisation that helped to conduct the research. Their participation in this study was supported through funding provided by Ipsen. Other authors have declared that no competing interests exist. Disclosures have been stated in the manuscript. This does not alter our adherence to PLOS ONE policies on sharing data and materials.

initially prescribed the 3-monthly or 6-monthly regimen remained on their original treatment, in contrast to only 38% for the 1-monthly formulation. Adherence analysis showed that 41.1% of 6-monthly injections were received without delay, compared with 67.9% for the 3-monthly and 77.3% for 1-monthly formulations. A large proportion of patients experienced delays of 14-27 days (32.0%, 33.4%, 54.2%) and over 27 days (39.6%, 48.3%, 46.6%) across the 1-, 3- and 6-monthly formulations respectively. The mean PDC ranged from 90-91% across the three formulation groups, with 89.9%, 84%, and 88.2% achieving ≥80% adherence for 3-monthly, 1-monthly, and 6-monthly respectively.

## Conclusions

This study revealed substantial and consistent dosing delays in LHRH agonist prescriptions across all formulations within primary care settings in England. These delays can negatively affect the control of PC, potentially hindering disease management for affected patients. Future research with a larger population, encompassing a larger cohort using the 6-monthly formulation, is essential for a comprehensive evaluation of the impact of LHRH agonist injection delays on PC progression.

## Introduction

Prostate cancer (PC) is the most frequently diagnosed cancer in men and is the second leading cause of male cancer deaths in England [1]. A mainstay of treatment for locally advanced and metastatic PC is androgen-deprivation therapy (ADT), specifically luteinising hormone-releasing hormone (LHRH) agonists which lower the level of testosterone [2]. LHRH agonists inhibit luteinising hormone secretion by inducing prolonged activation of LHRH receptors leading to desensitisation. This reduces testosterone production, which slows the growth and progression of PC cells [3]. ADT is also used in intermediate and high-risk localised PC, in the neoadjuvant and adjuvant setting prior to and after radical therapy is initiated, and as an adjuvant treatment to radical prostatectomy in patients with locally advanced PC at high risk of disease progression [4]. Castrate levels were previously described as < 50ng/dL, however a level of < 20 ng/dL is now recommended [5] and delays in LHRH agonist dosing may lead to an increase from these recommended castration levels of testosterone.

In England, most patients receive their first LHRH agonist injection in the secondary care setting and are then further treated in primary care. The routine receipt of LHRH agonist injections can be complex in the community setting, often consisting of multiple steps such as: decision to initiate in secondary care, communication to primary care, patient booking injection appointments, requesting the prescription, collecting their prescription from a community pharmacy, and taking the prescribed injection with them to their appointment at a primary care centre for administration. The standard of care and level of patients' involvement, as well as the steps that must be completed to receive an injection vary across England.

LHRH agonists used in PC treatment are commercially available in the UK as 1-, 3- or 6-monthly injection formulations. Some LHRH agonists are also available as every 28-day formulations, which will be referred to as 1-monthly in this study for simplicity. A study analysing data from the USA investigated the timeliness of LHRH agonist dosing by the different formulations (1, 3, 4 and 6-monthly) [6]. Results from the analysis suggested late injections, defined as receipt on any day after the pre-specified 28-, 84-, 112- or 168-day window, were common, with 84% (71,426) of injections received late. When an extended definition was used, defined as delay of four or more days after the pre-specified 28-, 84-, 112- or 168-day

window, late injections were less frequent, but more than one quarter (27%) (22,959) of injections were still received late [6]. With a lack of existing UK-specific data, our study assessed adherence to LHRH agonist treatment in England. We examined formulation-related adherence differences and its effect on overall survival. We also analysed how different formulations and adherence rates affected prostate specific antigen (PSA) changes over time. It is important to note that while this analysis assumes a likely causal relationship between variance in PSA and adherence schedule, such a connection may not always be definitive.

## Methods

Our retrospective cohort study used longitudinal primary care data drawn from the clinical management system used in GP practices and collated by the CPRD in a database named Aurum. The CPRD Aurum data is linked to secondary care data in the form of a reimbursement dataset called the Hospital Episode Statistics (HES) and death registrations collated by the Office for National Statistics (ONS). This study utilised anonymised structured data, preventing the authors with data access from identifying individual participants both during and after data collection.

Male patients aged 40 years and above, diagnosed with PC between 01 January 2007 and 31 December 2019 (study period [duration during which data was collected and analysed]: 01 April 2007 – 31 March 2020, index period [time in which patients received treatment]: 01 January 2007 – 31 December 2019) who had been prescribed an LHRH agonist injection, available in 1-, 3-, or 6-monthly formulations licensed in the UK within the primary care setting were included (Fig 1). The cut-off date for the inclusion criteria was set to prevent the effects of a change in the standard of care during the COVID-19 pandemic affecting the analysis. A minimum of two primary care prescriptions were required to allow the calculation of the proportion of days covered (PDC). Subjects were excluded from the study population on several criteria. These included having a single prescription for an LHRH agonist during

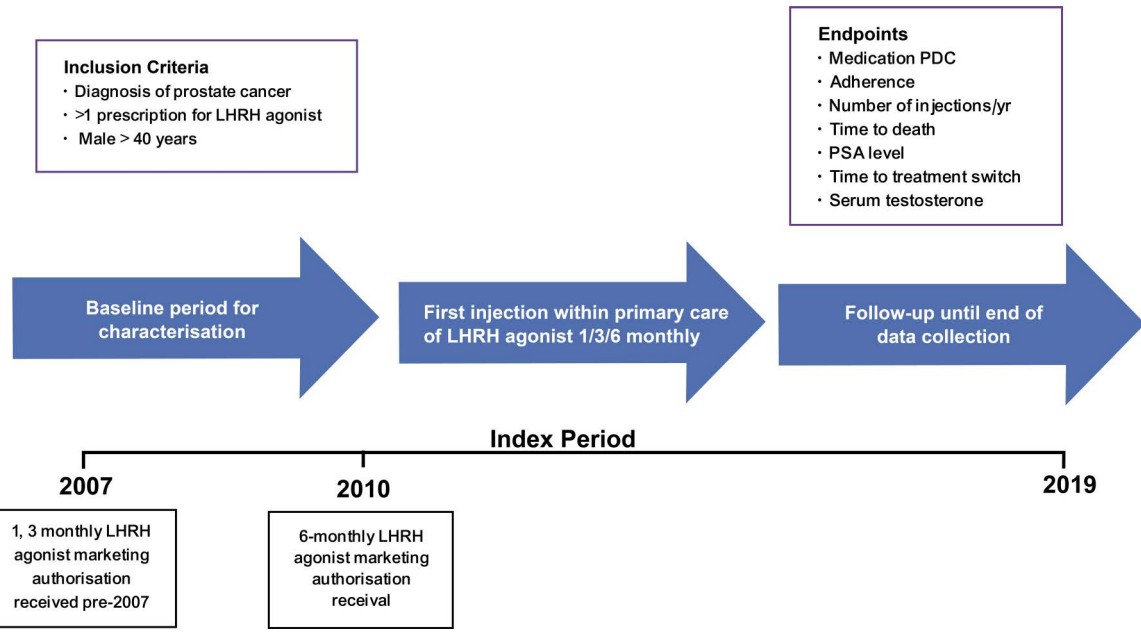

**Fig 1. Study design to meet the descriptive objectives.**

the study period or any prescription of an LHRH agonist or antagonist injection before the index date. Patients with less than six months of registration time at their CPRD contributing primary care practice before the index date or less than six months of follow-up time post index date were also excluded, as this limited the ability to identify switching. Finally, a prior diagnosis of cancer other than metastatic prostate cancer before an LHRH agonist injection prescription also resulted in exclusion. Patients were stratified by LHRH agonist formulation. Additionally, patients were further stratified if they remained on their initial formulation or switched to another formulation: from 1-monthly to 3- or 6-monthly, from 3-monthly to 1- or 6-monthly, and from 6-monthly to 1- or 3-monthly. Follow-up was conducted until the earliest occurrence of death, loss to follow-up in the dataset (i.e., last data collection for the primary care practice or patient transferred out of the practice) or the end of the index period.

The primary objectives of this study were to measure adherence to the 1-, 3- and 6-monthly LHRH agonist injection formulations through PDC, and to describe the number of delayed injections occurring < 4, 4-6, 7-13, 14-27 and ≥ 28 days beyond the pre-specified time window as per the summary of product characteristics for each of the 1-, 3- and 6-month formulations. The number of delayed injections stratified for the intervals < 4, 4-6, 7-13, 14-27 and ≥ 28 days are presented with absolute (and percentage) number of counts. Injections occurring with a delay of < 4 days were classified as "no delay". PDC was calculated by dividing the number of days between the date of receiving LHRH agonist injection to the subsequent LHRH agonist injection, by the number of days the LHRH agonist injection was intended to cover. When calculating PDC, it was assumed that prescriptions were for 28 days x the number of months the regimen is intended to cover + 3 days to allow for delays caused by weekends and public holidays. Prescription issue dates served as a proxy for adherence schedule, as the actual administration dates of LHRH agonists were not available in the CPRD dataset. This method is indicative of actual administration dates, although potentially underestimates delays in some patients.

A secondary objective of this study was to characterise patients by demographics, comorbidities, and time from diagnosis by LHRH formulation. Comorbid conditions at baseline were summarised using the Charlson Comorbidity Index (CCI) score and for specific pre-defined conditions: hypertension, high low-density lipoprotein (LDL), diabetes, obesity, atrial fibrillation, acute myocardial infarction, ischemic stroke, and heart failure. Description of the cause of death and overall survival for patients treated with LHRH agonist injections stratified by LHRH formulation was another secondary objective of this study. Kaplan Meier curves were generated to describe the median survival and survival probability for all patients and increasing PDC intervals < 80%, 80- < 85%, 85- < 90%, 90- < 95%, 95- < 100% and 100% were reported from 6 to 48 months at every 6-month interval. The final secondary objective of this study was to compare and describe PSA levels in patients that made up the study population stratified by formulation prescribed and adherence measured by PDC. Follow-up was restricted to one year, due to a lack of data availability.

Exploratory endpoints regarding PSA and testosterone distribution have been reported. Less than 6% of the whole cohort had a testosterone measurement recorded in CPRD dataset in the first year of follow up. Beyond the first year, the data became even more sparse and due to this missing data, no further analysis was conducted. Almost 50% of the whole cohort had at least one PSA measurement available in the CPRD dataset during follow up. For similar reasons as to testosterone distribution, we did not report past 1 year of follow-up due to the large volume of missing data.

The statistical analysis was descriptive therefore no formal statistical significance testing was performed. Descriptive statistics were presented as: number of available observations (N), number of missing values (missing), mean, SD, median, 1st and 3rd quartile and the range

(minimum, maximum) for continuous variables; the absolute and relative (percentage) numbers presented for categorical and discrete variables.

CPRD has ethics approval from the Health Research Authority to support research using anonymised patient data. Access to a suitable dataset to conduct this study was approved by the then ISAC (now RDG) on 05/11/2021 (protocol number 21 000622). The study adheres to ethical principles as outlined in the Declaration of Helsinki and the guidelines for Good Pharmacoepidemiology Practices from the International Society for Pharmacoepidemiology. It adhered to all local regulatory requirements applicable to non-interventional studies.

## Results

### Demographic characteristics

The study population comprised of 32,777 patients with PC treated with an LHRH agonist with at least 2 injections recorded during follow-up in primary care (Fig 2). Approximately 67% (21,910) of patients were initially prescribed the 3-monthly LHRH agonist injection, while 32% (10,365) were initially prescribed the 1-monthly formulation, and only 2% (502) received the 6-monthly formulation initially (Fig 2).

The overall mean age was 74.1 years (SD 8.7), with minor variations by formulation: patients on the 1-monthly, 3-monthly, and 6-monthly treatments had mean ages of 72.7 years (SD 8.3), 74.7 years (SD 8.8), and 74.5 years (SD 8.8), respectively (Table 1). Over 80% of patients had at least one comorbidity (based on number of patients with a record of comorbidities), with diagnosed hypertension being the most common (around 60% across all cohorts) (Table 1). Using the CCI, most patients (~40%) had scores of 1 or 2. Higher CCI scores were generally observed in patients on longer formulations (≥3: 9.5% for 1-monthly formulation versus 13.6% for 6-monthly formulation) (Table 1).

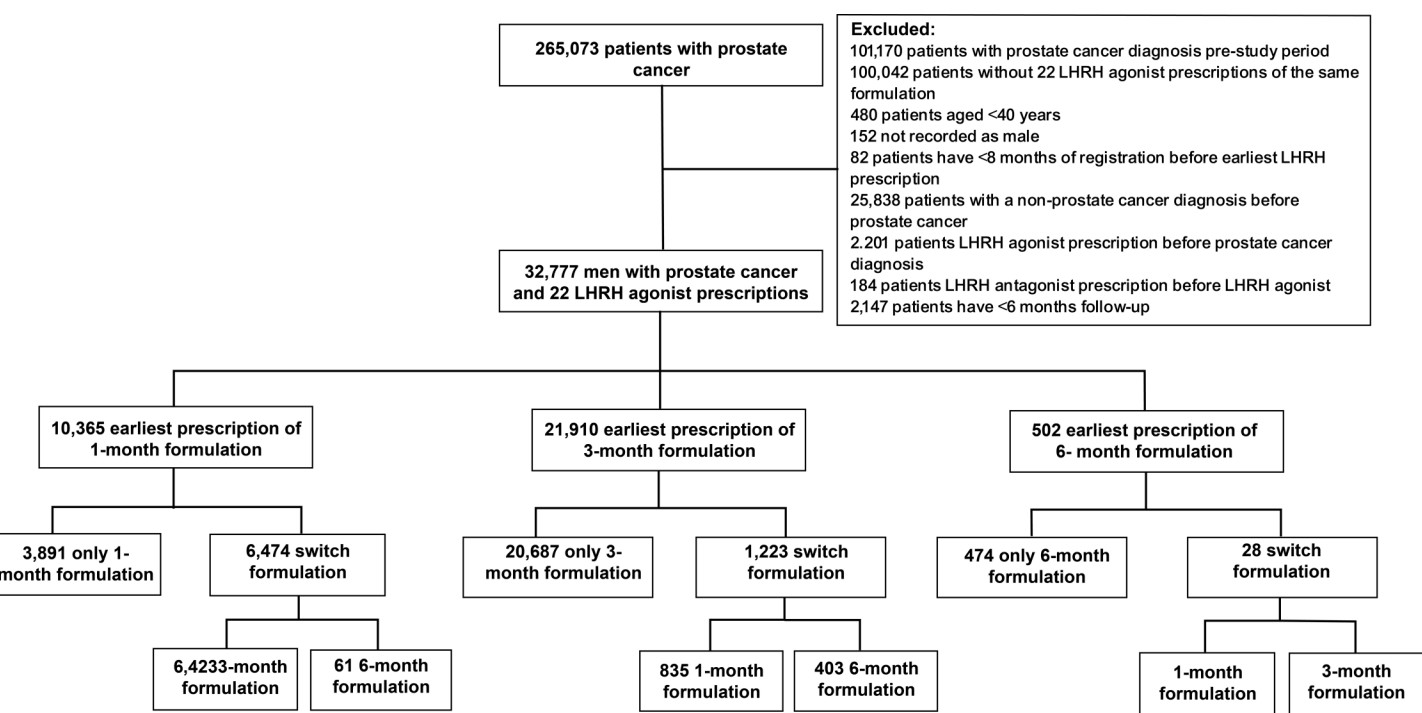

**Fig 2. Study population flow chart.** *When reporting data, CPRD policy is that no cell should contain <5 events.

**Table 1. Baseline characteristics table.**

| Metrics | | Whole study population | 1-month LHRH agonist injection | 3-month LHRH agonist injection | 6-month LHRH agonist injection |
|---|---|---|---|---|---|
| Total number of patients | | 32,777 | 10,365 | 21,910 | 502 |
| Total time in cohort (patient days) | | 53,371,173 | 19,621,778 | 33,184,728 | 564,667 |
| Follow up time (months) * | Mean | 54.3 | 63.1 | 50.5 | 37.5 |
| | Standard deviation | 37.2 | 40.4 | 35.2 | 18.1 |
| | Median | 45.5 | 57.1 | 41.3 | 36.0 |
| | Maximum | 158.1 | 157.4 | 158.1 | 104.0 |
| | Minimum | 6.0 | 6.0 | 6.0 | 6.8 |
| | Q1-Q3 | 22.8 – 79.07 | 27.7 – 94.5 | 21.37 – 72.37 | 22.8 – 49.62 |
| Age (years) | Mean | 74.1 | 72.7 | 74.7 | 74.5 |
| | Standard deviation | 8.7 | 8.3 | 8.8 | 8.8 |
| | Median | 74.0 | 73.0 | 75.0 | 75.0 |
| | Maximum | 100 | 100 | 100 | 97 |
| | Minimum | 40 | 40 | 41 | 48 |
| | Q1-Q3 | 68 to 80 | 67 to 78 | 69 to 81 | 68 to 81 |
| Age groups | <65 years | 4431 (13.5%) | 1659 (16%) | 2710 (12.4%) | 62 (12.4%) |
| | 65-69 years | 5195 (15.9%) | 1874 (18.1%) | 3239 (14.8%) | 82 (16.3%) |
| | 70-74 years | 7141 (21.8%) | 2520 (24.3%) | 4521 (20.6%) | 100 (19.9%) |
| | 75-79 years | 7307 (22.3%) | 2300 (22.2%) | 4897 (22.4%) | 110 (21.9%) |
| | ≥80 years | 8703 (26.6%) | 2012 (19.4%) | 6543 (29.9%) | 148 (29.5%) |
| Charlson comorbidity index score | Number of patients with available data** | 16829 (51.3%) | 4962 (47.9%) | 11601 (53%) | 266 (53%) |
| | 0** | 0 (0%) | 0 (0%) | 0 (0%) | 0 (0%) |
| | 1-2 | 13230 (40.4%) | 3981 (38.4%) | 9051 (41.3%) | 198 (39.4%) |
| | 3-4 | 2916 (8.9%) | 806 (7.8%) | 2058 (9.4%) | 52 (10.4%) |
| | 5-9 | 678 (2.1%) | 174 (1.7%) | 489 (2.2%) | 15 (3%) |
| | 10-14 | 5 (0%) | <5[†] | <5[†] | <5[†] |
| | 15+ | 0 (0%) | 0 (0%) | 0 (0%) | 0 (0%) |
| | Number of patients without available data | 15948 (48.7%) | 5403 (52.1%) | 10309 (47.1%) | 236 (47%) |
| Comorbidities | Number of patients with record of comorbidities** | 27091 (82.7%) | 8571 (82.7%) | 18102 (82.6%) | 418 (83.3%) |
| | Diagnosed hypertension | 19438 (59.3%) | 5957 (57.5%) | 13170 (60.1%) | 311 (62%) |
| | High LDL | 17080 (52.1%) | 5732 (55.3%) | 11089 (50.6%) | 259 (51.6%) |
| | Diabetes | 6041 (18.4%) | 1784 (17.3%) | 4149 (18.9%) | 98 (19.5%) |
| | Obesity | 6993 (21.3%) | 2183 (21.1%) | 4706 (21.5%) | 104 (20.7%) |
| | Atrial fibrillation | 4015 (12.3%) | 1082 (10.4%) | 2878 (13.1%) | 55 (11%) |
| | Acute myocardial infarction | 2791 (8.5%) | 796 (7.7%) | 1958 (8.9%) | 37 (7.4%) |
| | Ischemic stroke | 953 (2.9%) | 241 (2.3%) | 705 (3.2%) | 7 (1.4%) |
| | Heart failure | 1770 (5.4%) | 459 (4.4%) | 1279 (5.8%) | 32 (6.4%) |
| | Number of patients with no record of comorbidities | 5686 (17.4%) | 1794 (17.3%) | 3808 (17.4%) | 84 (16.7%) |

*(Continued)*

**Table 1.** (Continued)

| Metrics | | Whole study population | 1-month LHRH agonist injection | 3-month LHRH agonist injection | 6-month LHRH agonist injection |
|---|---|---|---|---|---|
| Geographical region | East of England | 1769 (5.4%) | 402 (3.9%) | 1256 (5.7%) | 111 (22.1%) |
| | South East | 3011 (9.2%) | 1350 (13%) | 1657 (7.6%) | <5† |
| | London | 4676 (14.3%) | 2044 (19.7%) | 2620 (12%) | 12 (2.4%) |
| | South West | 4045 (12.3%) | 1237 (11.9%) | 2758 (12.6%) | 50 (10%) |
| | East Midlands | 964 (2.9%) | 244 (2.4%) | 714 (3.3%) | 6 (1.2%) |
| | Yorkshire and Humber | 1086 (3.3%) | 293 (2.8%) | 792 (3.6%) | <5† |
| | North East | 960 (2.9%) | 109 (1.1%) | 808 (3.7%) | 43 (8.6%) |
| | North West | 5996 (18.3%) | 2207 (21.3%) | 3742 (17.1%) | 47 (9.4%) |
| | West Midlands | 6475 (19.8%) | 1524 (14.7%) | 4976 (21.9%) | 155 (30.9%) |
| | South Central | 3795 (11.6%) | 955 (9.2%) | 2767 (12.6%) | 73 (14.5%) |
| Ethnicity | White | 26720 (81.5%) | 8386 (80.9%) | 17894 (81.7%) | 440 (87.6%) |
| | Mixed | 65 (0.2%) | 17 (0.2%) | 47 (0.2%) | <5† |
| | Asian or Asian British | 454 (1.4%) | 157 (1.5%) | 284 (1.3%) | 13 (2.6%) |
| | Black or Black British | 1175 (3.6%) | 461 (4.4%) | 706 (3.2%) | 8 (1.6%) |
| | Chinese or Other Group | 36 (0.1%) | 12 (0.1%) | 23 (0.1%) | <5† |
| | Number of patients with missing data | 4327 (13.2%) | 1332 (12.9%) | 2956 (13.5%) | 39 (7.8%) |
| Index of multiple deprivation quintile | Number of patients with available data | 32745 (99.9%) | 10349 (99.9%) | 21895 (99.9%) | 501 (99.8%) |
| | 1 (least deprived) | 8222 (25.1%) | 2587 (25%) | 5507 (25.1%) | 128 (25.5%) |
| | 2 | 7620 (23.3%) | 2484 (24%) | 5018 (22.9%) | 118 (23.5%) |
| | 3 | 6597 (20.1%) | 2136 (20.6%) | 4355 (19.9%) | 106 (21.1%) |
| | 4 | 5550 (16.9%) | 1784 (17.2%) | 3675 (16.8%) | 91 (18.1%) |
| | 5 (most deprived) | 4756 (14.5%) | 1358 (13.1%) | 3340 (15.2%) | 58 (11.6%) |
| | Number of patients with missing data | 32 (0.1%) | 16 (0.2%) | 15 (0.1%) | <5† |

Definitions: LHRH = Luteinising hormone-releasing hormone; Q1-Q3 = First Quartile-Third Quartile.

*Followed from the date of first prescription until earliest event of death, transfer out of general practice, last record collection from general practice no further Hospital Episode Statistics entry, or 365 days with no LHRH agonist injection prescription post-date of missed injection.

**As data are drawn from medical records, the absence of a comorbidity implies the condition is not present rather than the data not being available. As the study population have PC, a condition included in the CCI score no patients can have a score of zero.

Patients initially treated with the 3-monthly regimen were least likely to switch to a different formulation (1-monthly or 6-monthly), with 94% (20,687) remaining on their original treatment. Similarly, 94% (474) of patients on the 6-monthly regimen also did not switch treatments, although the cohort size was small in the time period covered. In comparison, only 38% (3,891) of patients on the 1-monthly treatment stayed unchanged. Most patients (20% [6,447/32,777]) were switched from their initially prescribed formulation to the 3-monthly formulation rather than the 1-monthly (3% [839/32,777]) or 6-monthly (1% [464/32,777]).

The overall mean follow-up period for the whole study population was 54.3 ($\approx$ 4.5 years) months, SD 37.2. Patients treated with the 6-monthly formulation had a shorter follow-up period (mean of 37.5 months, SD 18.1) in keeping with when this preparation came into use.

In contrast, longer follow-ups were observed for the 1-monthly formulation (mean of 63.1 months, SD 40.4) and the 3-monthly formulation (mean of 50.5 months, SD 35.2). The predominant ethnicity within the study population was white, constituting 81.5% of the overall participants.

Based on calculations using ONS data (Table 2), the prevalence of black or black British patients in the study population (0.40% [1,175]) was nearly twice that of white patients (0.24% [26,720]). This finding aligns with existing research indicating that black men have a higher likelihood of developing prostate cancer compared with individuals from other racial backgrounds [7].

## Primary endpoint - patient adherence measured by delayed LHRH agonist injections

Data presented show that the proportion of injections received without a delay (within 4 days of date receipt was due) was 41.1% for 6-monthly, 67.9% for the 3-monthly and 77.3% for 1-monthly LHRH agonist formulation (Fig 3). However, a large proportion of patients experienced at least one injection delay of 14-27 days (32.0% for 1-monthly, 33.4% for the 3-monthly and 54.2% for 6-monthly LHRH agonist formulation) (Fig 3). Furthermore, 39.6%, 48.3% and 46.6% of patients in the 1-, 3-, and 6-monthly LHRH agonist formulation cohorts experienced at least one injection delay of over 27 days (Fig 3).

## Primary endpoint - patient adherence measured by proportion of days covered (PDC)

The mean PDC was similar across all formulation groups. Both the 3- and 6-monthly cohorts achieved a PDC of 91% and the 1-monthly cohort had a PDC of 90% (Table 3). The 1-monthly cohort contained 18.5% (1,919) of patients who achieved a 100% PDC, whereas among the 3- and 6-monthly cohorts this percentage decreased to 11.6% (2,537) and 6% (30), respectively (Table 3). When using a PDC of ≥80% as a cut-off to define adherence, 89.9% of the 6-monthly cohort were adherent versus 84% and 88.2% for the 1- and 3-monthly cohorts, respectively. Table 3 contains additional PDC interval measurements for the 1-, 3-, and 6-monthly formulations.

## Secondary endpoint - overall survival analysis

The (Fig 4) median overall survival (OS) was 111 months (95% CI = 107-114 months) for the 1-monthly formulation, 72 months (95% CI = 71–74 months) for the 3-monthly and 58 months (95% CI = 55-66 months) for the 6-monthly (Fig 4). It should be noted that a large

**Table 2. Ethnicity prevalence.**

| Metrics | | Whole study population | Number of makes >40 years in England | Whole study population ethnicity prevalence* | 1-month LHRH agonist injection | 3-month LHRH agonist injection | 6-month LHRH agonist injection |
|---|---|---|---|---|---|---|---|
| Ethnicity | White | 26,720 (81.5%) | 11,337,776 | 0.24% | 8,386 (80.9%) | 17,894 (81.7%) | 440 (87.6%) |
| | Mixed | 65 (0.2%) | 98,445 | 0.07% | 17 (0.2%) | 47 (0.2%) | 1 (0.2%) |
| | Asian or Asian British | 454 (1.4%) | 427,854 | 0.11% | 157 (1.5%) | 284 (1.3%) | 13 (2.6%) |
| | Black or Black British | 1,175 (3.6%) | 297,298 | 0.40% | 461 (4.4%) | 706 (3.2%) | 8 (1.6%) |
| | Chinese or Other Group | 36 (0.1%) | 253,279 | 0.01% | 12 (0.1%) | 23 (0.1%) | 1 (0.2%) |

*Calculations based on 2011 Office of National Statistics data for males > 40 years in England.

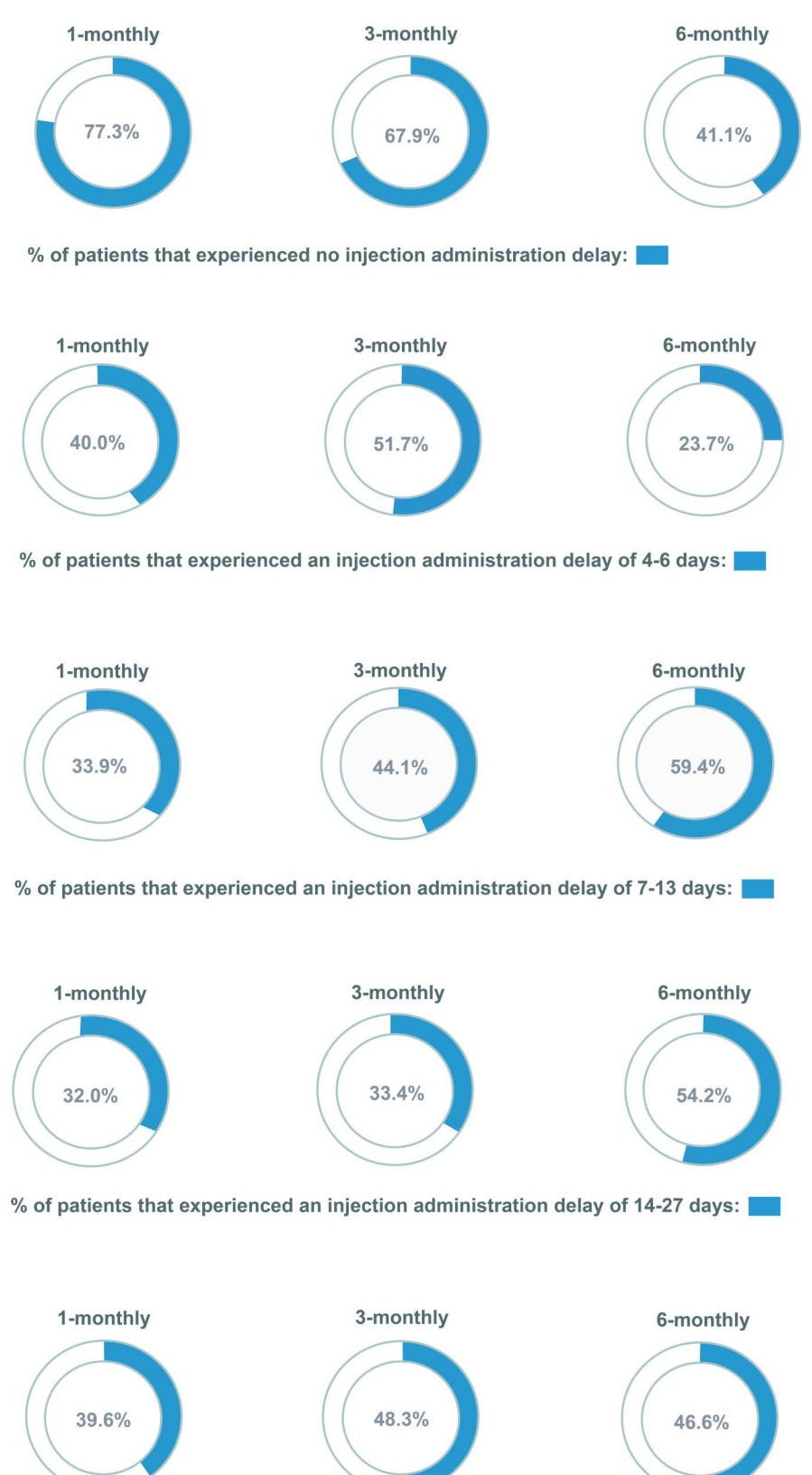

**Fig 3. Patient adherence measured by delayed LHRH agonist injection statistics.**

**Table 3. Patient adherence measured by proportion of days covered.**

| Metrics | | Whole study population | 1-monthly LHRH agonist cohort | 3-monthly LHRH agonist cohort | 6-monthly LHRH agonist cohort |
|---|---|---|---|---|---|
| Proportion of days covered (PDC) | Mean | 0.91 | 0.90 | 0.91 | 0.91 |
| | Standard deviation | 0.11 | 0.13 | 0.10 | 0.08 |
| | Median | 0.94 | 0.94 | 0.94 | 0.94 |
| | Maximum | 1.00 | 1.00 | 1.00 | 1.00 |
| | Minimum | 0.16 | 0.16 | 0.23 | 0.46 |
| | Q1-Q3 | 0.87 – 0.98 | 0.86 – 0.99 | 0.87 – 0.98 | 0.9 – 0.96 |
| PDC by intervals | <80% | 4288 (13.1%) | 1653 (16.0%) | 2584 (11.8%) | 51 (10.2%) |
| | 80% - <85% | 2559 (7.8%) | 813 (7.8%) | 1720 (7.9%) | 26 (5.2%) |
| | 85% - <90% | 4103 (12.5%) | 1297 (12.5%) | 2748 (12.5%) | 58 (11.6%) |
| | 90% - <95% | 6888 (21.0%) | 1885 (18.2%) | 4799 (21.9%) | 204 (40.6%) |
| | 95% - <100% | 10453 (31.9%) | 2798 (27.0%) | 7522 (34.3%) | 133 (26.5%) |
| | 100% | 4486 (13.7%) | 1919 (18.5%) | 2537 (11.6%) | 30 (6.0%) |
| Total number of injections | | 368,803 | 128,033 | 238,232 | 2,538 |

Definitions: LHRH=Luteinising hormone-releasing hormone; PDC=Proportion of days covered; Q1-Q3 = First Quartile-Third Quartile.

proportion (62%) of patients initially commenced on a 1-monthly formulation were then subsequently switched to a longer duration formulation, so the data represents mixed dose regimens. When stratifying the Kaplan Meier curves by PDC, a survival probability of at least 60% is observed for the 1- and 3-monthly formulations at 48 months regardless of PDC. Furthermore, the analysis revealed that within the 1-monthly cohort, patients who achieved a 100% PDC demonstrated an increased survival probability (Fig 4). Due to the small sample size in the 6-monthly cohort, there was no stratification in survival by PDC level.

## Secondary endpoint - cause of death

Among the patients who died during follow-up, PC was considered the highest primary (69.6%) and secondary (16.0%) cause of death (Figs 5 and 6). Prostate cancer being recorded as a secondary cause of death is in accordance with the standard practice on UK death certificates. In such cases, the primary cause of death is the condition that directly leads to the fatality, while the secondary cause may represent an underlying or contributing factor.

## Discussion

Most patients with PC in the study were initially started on a 3-monthly LHRH agonist formulation. Patients switching formulations mostly switched from 1-monthly to 3-monthly. The vast majority (94%) of patients initially prescribed a 3-monthly or 6-monthly formulation did not switch, which suggests that these formulations are well-tolerated and practical for healthcare services to deliver. Formulation usage and patient characteristics were broadly consistent with data from the USA [6].

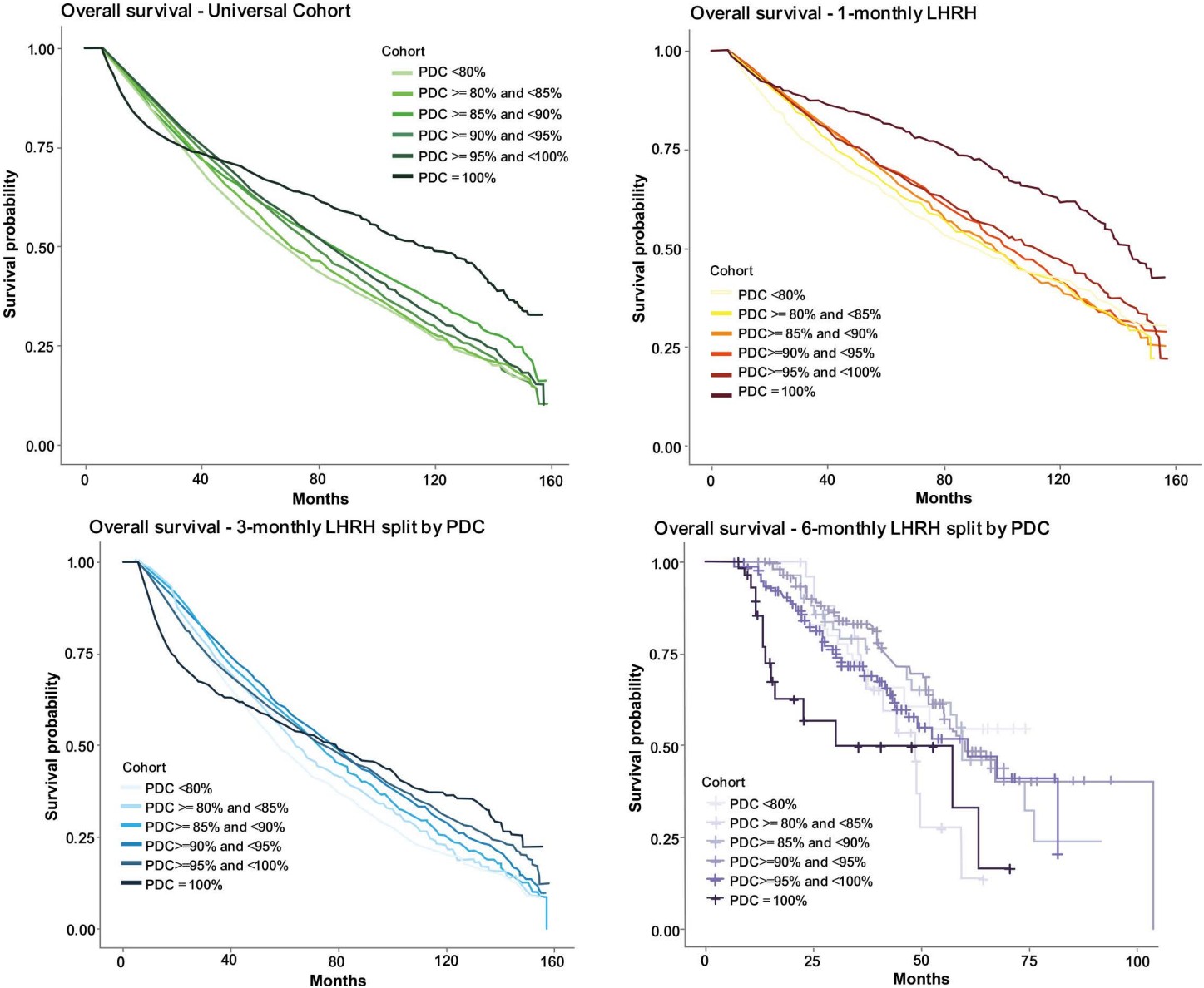

**Fig 4. Kaplan Meier curves for the overall survival estimates for each LHRH agonist formulation, further stratified for PDC coverage.**

A very high proportion of patients were exposed to delays (defined as a delay greater than or equal to 4 days) in the administration of their LHRH agonist injections, across all formulation cohorts. Overall, 45.5% (14,925) of patients in the study population experienced at least one injection delay exceeding 27 days, more specifically, 39.6% (4,106), 48.3% (10,585), and 46.6% (234) of patients in the 1-monthly, 3-monthly, and 6-monthly cohorts, respectively. These delays may reasonably be expected to result in a reduction in testosterone suppression in some cases. While the absence of testosterone measurements in the CPRD database means we have not been able to directly confirm this, a potential for avoidable disease progression seems to exist. In addition, the methodology used is likely to have resulted in a conservative estimate of treatment delays because it is based on the point of prescription rather than administration.

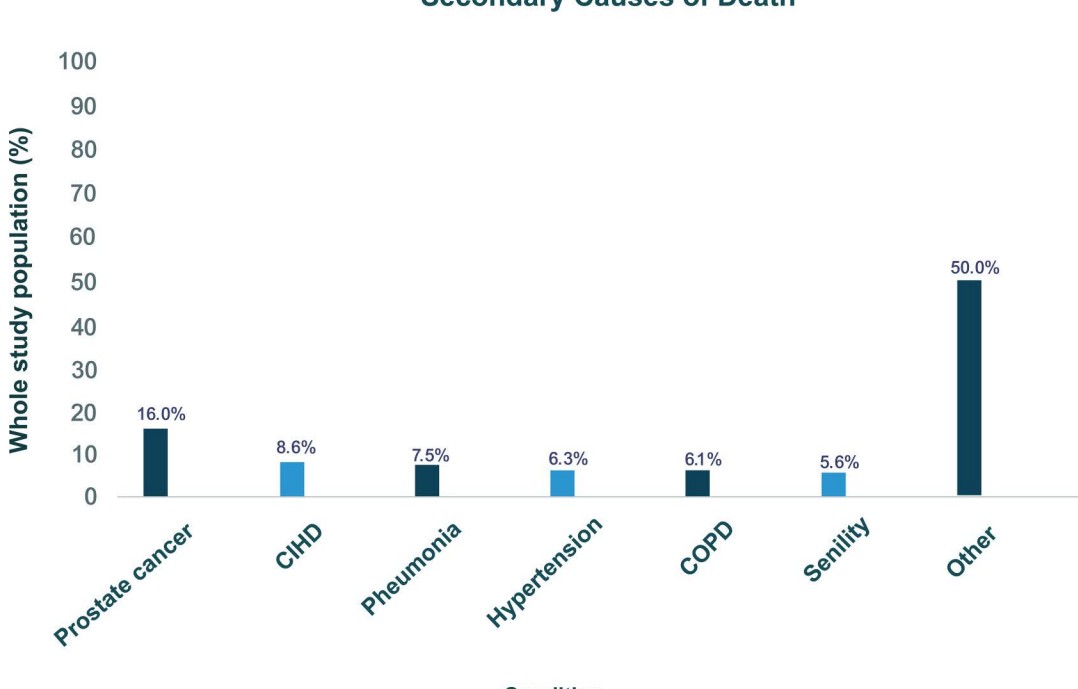

**Fig 5. Graph showing the primary causes of death in descending order for the top six conditions.** Definitions: CIH-D=chronic ischaemic heart disease; MI=myocardial infarction; MN=malignant neoplasm.

**Fig 6. Graph showing the secondary causes of death in descending order for the top 6 conditions.** Definitions: CIH-D=chronic ischaemic heart disease, COPD=chronic obstructive pulmonary disease.

A possible explanation for delays in the 3- and 6-monthly cohorts could be that prescriptions for LHRH agonist injections are based on calendar time rather than the intervals recommended by manufacturers. This deviation may be influenced by factors such as the limited availability of healthcare professionals, including general practitioners (GPs) and nurses. This would not have affected delays in the 1-monthly formulation cohort as, delays of 'less than 4 days' were not classified as a 'delay' in this study. Primary care practices determine how far in advance appointments can be made and scheduling appointments six months ahead is logistically difficult. Furthermore, based on our own clinical practice experiences, there may be a perception that the clinical consequences are less significant for treatment delays with the longer-acting formulations as the medicine and its effects are perceived to persist in the system for longer. Additionally, we postulate that higher patient comorbidity could contribute to the increased delays in administering LHRH agonist injections. The study results align with a similar analysis in the USA [6], which found more instances of late dosing for longer-acting formulations (6-monthly: 95% late, 1-monthly: 73% late).

Overall, findings in relation to the 6-monthly formulation should be interpreted with caution due to the small number of patients who received this formulation. The relatively low adoption of the 6-monthly formulation can be attributed to its delayed availability for use in the NHS compared with the other formulations (although contemporary practice has evolved since the COVID-19 pandemic) [8]. Adherence rates were similar across formulations, with some minor differences for those on a 6-monthly formulation. A lower proportion of patients initiated on a 6-monthly formulation achieved a 100% treatment coverage when compared with those initiated on 1- and 3-monthly formulations. However, among patients in the 6-monthly group, 89.9% achieved a PDC of ≥ 80%, compared with 84% and 88.2% for the 1- and 3-monthly groups respectively, which shows that these patients achieved the highest levels of adherence to their prescribed therapy plan.

It is important to note that the data available for testosterone and PSA measurements were limited, with less than 6% of individuals having testosterone measurements recorded in primary care, and only 50% having at least one PSA measurement during follow-up. PSA and testosterone measurements are crucial for monitoring and managing PC, and the lack of recorded data on these measurements poses a significant challenge. One possible explanation for the low recorded testosterone and PSA measurements in the study population is that data from CPRD is limited to primary care, and that these data are only available in secondary care electronic health records where prostate cancer is typically diagnosed and managed. Additionally, it is worth noting that this data is not available in the HES database, which does contain secondary care data. However, the HES database is more focused on patient outcomes for reimbursement purposes rather than laboratory results, and consequently lacks information on testosterone and PSA measurements. Another explanation may be attributed to factors such as the cost of testing or a potential lack of clinician awareness regarding the necessity to assess these metrics.

The primary consideration when evaluating the overall survival is the limited sample size and shorter follow-up period in the 6-monthly cohort. When comparing patients on the 6- and 3-monthly formulation, there was no statistically significant difference in overall survival. It should also be noted that these are 'all cause' deaths and not PC specific, of which there was a lower proportion in the 6-monthly formulation cohort. Additionally, competing risks have not been accounted for. The study found statistically significant differences in overall survival (OS) between the 1-monthly and 3-monthly formulations, with patients receiving the 1-monthly formulation exhibiting a higher median OS than those receiving the 3-monthly formulation. However, it's important to note that many patients initially started on the 1-monthly formulation were later switched to longer-duration formulations. Based on our experience, this change is

primarily due to hospital discharge policies, which mandate a 7-day medicines supply upon discharge to lower treatment costs incurred by the hospital. Since the smallest and cheapest pack size available is the 1-monthly formulation, follow-up care is often transferred to primary care where GPs, considering patient preferences and their workload, may choose to prescribe 3-month or 6-month formulations. Upon stratifying the data by PDC, those patients receiving the 1-monthly formulation, who exhibited a 100% coverage rate, displayed an increased probability of survival (after approximately 30 months). However, it is important to highlight that a significant portion of patients who were initially started on the 1-monthly formulation later switched to a longer duration regimen, resulting in a dataset that contains mixed dosing regimens. Studies have indicated that discontinuing lower duration LHRH agonist injections could lead to a quicker regeneration of testosterone levels compared with when stopping longer duration formulations [9]. This finding offers a potential explanation for fewer treatment delays and the increased sensitivity of patients on the 1-monthly formulation to treatment delays.

In line with previous research [10], this study, along with data from the Office for National Statistics (ONS), demonstrates that the prevalence of PC among patients from a black or black British background was approximately twice as high compared with white patients, emphasizing findings across multiple studies and geographies.

## Limitations

A significant limitation of this study pertains to the assessment of the 6-monthly LHRH agonist formulation. Unlike the other formulations examined, the 6-monthly formulation product had a relatively shorter period of formulary availability within the UK based NHS during the study period [11]. Consequently, the sample size of patients receiving the 6-monthly formulation was notably smaller in comparison to the 1-monthly and 3-monthly cohorts. This lack of data limits our ability to draw valid conclusions regarding the long-term treatment outcomes and overall survival of patients on the 6-monthly formulation compared with the other formulations.

Another limitation is that the "dates" captured in the CPRD database relate to the date the prescription was issued, rather than the administration of injection itself. As such, the administration date was inferred from the prescription issue date. As prescriptions can only be administered after issue and collection, this would mean that the dosing delays could be significantly underestimated in this analysis but cannot be shorter. Furthermore, it is important to note that the CPRD database does not provide reasons behind the delays, which is an additional limitation of this study. Without access to the specific reasons for these delays, the analysis is limited in its ability to capture the full spectrum of factors influencing treatment adherence.

In addition, a significant number of patients will have received their first prescription in secondary care, and this cannot be captured in the available CPRD-HES linked database as the secondary care component only covers medical activity and does not contain secondary care prescription data. The impact of this would mean that for any patients first prescribed a LHRH agonist in secondary care, the first prescription captured in the primary care is not their genuine treatment start. Although this could account for a higher rate of switching with primary care prescribers preferring a longer interval to reduce the number of clinic appointments.

Neither the administration of other treatments, PC-related surgery, nor the time from diagnosis to treatment were considered in this analysis. There may be imbalance between locally advanced and metastatic disease across the formulation groups which cannot be distinguished in the data.

## Concluding statements

Overall, the study identified significant and systemic dosing delays across all formulations of LHRH agonist prescriptions in primary care settings across England, these have the potential

to significantly adversely affect patients with PC. Further studies with a larger population, including a larger cohort receiving the 6-monthly formulation will be necessary to accurately assess the impact of delays on PC progression.

## Acknowledgements

We would like to express our gratitude to Ravina Barrett for her assistance in reviewing this paper. Her expertise and constructive criticism have been instrumental in helping create the final version of this paper.

## Author contributions

**Formal analysis:** Ian Sayers, Sara Joao Carvalho, Jennifer Davidson, Naomi Elster, Rakesh Heer, Kate Higgs, Andrew Nolan, Jelena Sassmann.

**Supervision:** Ian Sayers, Sara Joao Carvalho, Jennifer Davidson, Naomi Elster, Rakesh Heer, Kate Higgs, Andrew Nolan, Jelena Sassmann.

**Writing – original draft:** Mohammad Raja.

**Writing – review & editing:** Ian Sayers, Sara Joao Carvalho, Jennifer Davidson, Naomi Elster, Rakesh Heer, Kate Higgs, Andrew Nolan, Jelena Sassmann.

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
