## [Decision Letter · Decision Letter 0]

15 Jul 2024

PONE-D-24-14050A retrospective cohort study assessing medication coverage in patients with prostate cancer prescribed luteinizing hormone releasing hormone (LHRH) agonists in EnglandPLOS ONE

Dear Dr. Sayers,

Thank you for submitting your manuscript to PLOS ONE. After careful consideration, we feel that it has merit but does not fully meet PLOS ONE’s publication criteria as it currently stands. Therefore, we invite you to submit a revised version of the manuscript that addresses the points raised during the review process.

According to the review reports, your study is interesting and well-focused. There are no major issues for revision. However, some minor concerns raised by Reviewer 2 should be addressed in this revision.

We look forward to receiving your revised manuscript.

Kind regards,

Wen-Wei Sung, M.D., Ph.D.

Academic Editor

PLOS ONE

Journal Requirements:

"Sara Joao Carvalho and Jennifer Davidson are employed by CorEvitas, an organisation that helped to conduct the research. Their participation in this study was supported through funding provided by Ipsen."

"Sara Joao Carvalho and Jennifer Davidson are employed by CorEvitas, an organisation that helped to conduct the research. Their participation in this study was supported through funding provided by Ipsen.

Other authors have declared that no competing interests exist. 

Disclosures have been stated in the manuscript."

We note that one or more of the authors are employed by a commercial company: Ipsen

A. Please provide an amended Funding Statement declaring this commercial affiliation, as well as a statement regarding the Role of Funders in your study. If the funding organization did not play a role in the study design, data collection and analysis, decision to publish, or preparation of the manuscript and only provided financial support in the form of authors' salaries and/or research materials, please review your statements relating to the author contributions, and ensure you have specifically and accurately indicated the role(s) that these authors had in your study. You can update author roles in the Author Contributions section of the online submission form.

B. Please also provide an updated Competing Interests Statement declaring this commercial affiliation along with any other relevant declarations relating to employment, consultancy, patents, products in development, or marketed products, etc.  

Within your Competing Interests Statement, please confirm that this commercial affiliation does not alter your adherence to all PLOS ONE policies on sharing data and materials by including the following statement: ""This does not alter our adherence to  PLOS ONE policies on sharing data and materials.” (as detailed online in our guide for authors http://journals.plos.org/plosone/s/competing-interests) . If this adherence statement is not accurate and  there are restrictions on sharing of data and/or materials, please state these. Please note that we cannot proceed with consideration of your article until this information has been declared.

5. We note that your Data Availability Statement is currently as follows: All relevant data are within the manuscript and its Supporting Information files.

7. Please include a separate caption for each figure in your manuscript.

8. Please include your tables as part of your main manuscript and remove the individual files. Please note that supplementary tables (should remain/ be uploaded) as separate ""supporting information"" files

Reviewers' comments:

Reviewer's Responses to Questions

**Comments to the Author**

1. Is the manuscript technically sound, and do the data support the conclusions?

Reviewer #1: Yes

Reviewer #2: Yes

2. Has the statistical analysis been performed appropriately and rigorously? 

Reviewer #1: Yes

Reviewer #2: Yes

3. Have the authors made all data underlying the findings in their manuscript fully available?

Reviewer #1: Yes

Reviewer #2: Yes

4. Is the manuscript presented in an intelligible fashion and written in standard English?

Reviewer #1: Yes

Reviewer #2: Yes

5. Review Comments to the Author

Reviewer #1: This manuscript assess adherence to LHRH agonist treatment for prostate cancer in England and revealed substantial and consistent dosing delays in LHRH agonist prescriptions across all formulations within primary care settings in England. These delays can negatively affect the control of PC, potentially hindering disease management for affected patients. It describes a technically sound piece of scientific research with data that supports the conclusions. The paper is well written and the findings are of interest to clinicians.

Reviewer #2: This article is good, there is new idea for treatment of androgen deprivation therapy (ADT) with LHRH agonist for advanced prostate cancer (PC), so we have more variation choice treatment of the Prostate cancer management

6. PLOS authors have the option to publish the peer review history of their article (what does this mean? ). If published, this will include your full peer review and any attached files.

**Do you want your identity to be public for this peer review?** For information about this choice, including consent withdrawal, please see our Privacy Policy .

Reviewer #1: No

Reviewer #2: No

---

## [Author Response · Author response to Decision Letter 1]

15 Nov 2024

1. Is there any inclusion and exclusion criteria about inj LHRH agonist 1, 3, and 6 months?

• Response: We appreciate the reviewer's comment. The inclusion criteria had already been mentioned in the methodology section. In response to this comment, we have now added the exclusion criteria as well, which can be found in the revised version of the methodology section.

Location in manuscript: Line 169.

2. Is there any reasoning about why the injections are delayed? (Perhaps based on patients’ condition, healthcare system, or anything else?)

• Response: The methodology section of the manuscript does not originally address reasons for delayed injections. However, we have explored this issue in greater depth in the discussion section of the paper, where we have added a detailed examination of potential reasons for delays, including patient condition and healthcare system factors.

Location in manuscript: Line 365.

3. Does patients' comorbidity affect the delayed injections?

• Response: We have expanded the discussion to now include an explanation of how comorbidities can influence the timing of LHRH agonist injections. This additional detail should clarify the potential role of comorbid conditions in contributing to injection delays.

Location in manuscript: Line 375.

4. How to determine prostate cancer as a primary cause of death if patients also have comorbidities?

• Response: The primary cause of death is determined by the death certificate, which serves as the definitive and legally binding document used for clinical research. As such, in this study, the cause of death was based on the death certificate as the most reliable source of information.

Location in manuscript: No specific reference, general clarification.

Additional Revisions and Compliance with PLOS ONE Guidelines:

1. Ensured adherence to PLOS ONE style requirements:

We have thoroughly reviewed and revised the manuscript to ensure full compliance with PLOS ONE style requirements, including formatting and citation guidelines.

2. Amended the role of funder statement:

We have revised the role of the funder statement, which is now included appropriately in the manuscript.

3. Amended the funding statement:

The funding statement has been updated to declare the commercial affiliation of Ipsen authors as per PLOS ONE’s requirements.

4. Updated competing interest statement:

We have amended the competing interest statement to clarify that "This does not alter our adherence to PLOS ONE policies on sharing data and materials."

Data Sharing Request Response:

• Response to request for raw data:

We have reviewed previous PLOS ONE publications that use data from CPRD and HES. Our data sharing statement has been updated to align with CPRD’s requirements and is consistent with the guidelines followed by previously accepted manuscripts:

Data are available upon request from the Clinical Practice Research Datalink (CPRD). However, access to this data requires the purchase of a license, and the terms of this license do not permit the authors to make the data publicly available to all. Licenses are available directly from CPRD (http://www.cprd.com): The Clinical Practice Research Datalink Group, The Medicines and Healthcare products Regulatory Agency, 10 South Colonnade, Canary Wharf, London E14 4PU.

Additional Changes Made to the Manuscript:

1. Moved the ethics statement to the methods section:

As requested, the ethics statement has now been moved to the methods section of the manuscript for proper placement.

2. Provided separate captions for each figure:

Each figure now has its own distinct caption, which is in accordance with PLOS ONE guidance.

3. Tables and figures:

The tables have been included within the main manuscript, and all figures have been submitted separately, as specified by PLOS ONE.

4. Reviewed reference list:

The reference list has been thoroughly reviewed to ensure that it is complete, correct, and fully adheres to PLOS ONE formatting requirements.

---

## [Editor Report · Decision Letter 1]

22 Nov 2024

A retrospective cohort study assessing medication coverage in patients with prostate cancer prescribed luteinizing hormone releasing hormone (LHRH) agonists in England

PONE-D-24-14050R1

Dear Dr. Sayers,

We’re pleased to inform you that your manuscript has been judged scientifically suitable for publication and will be formally accepted for publication once it meets all outstanding technical requirements.

Kind regards,

Wen-Wei Sung, M.D., Ph.D.

Academic Editor

PLOS ONE
---

## [Editor Report · Acceptance letter]

PONE-D-24-14050R1

PLOS ONE

Dear Dr. Sayers,

I'm pleased to inform you that your manuscript has been deemed suitable for publication in PLOS ONE. Congratulations! Your manuscript is now being handed over to our production team.

Kind regards,

on behalf of

Dr. Wen-Wei Sung

Academic Editor

PLOS ONE